# From Surface to Viscera: 3D Estimation of Internal Anatomy from Body Surface Point Clouds

**Salih Furkan Atici**[1] ⬤              SALIH.ATICI@UNI-LUEBECK.EDU

**Eytan Kats**[1] ⬤                  EYTAN.KATS@UNI-LUEBECK.EDU

**Daniel Mensing**[2] ⬤            DANIEL.MENSING@MEVIS.FRAUNHOFER.DE

**Mattias P Heinrich**[1] ⬤           MATTIAS.HEINRICH@UNI-LUEBECK.EDU

[1] *Institute of Medical Informatics, University of Luebeck, Luebeck, Germany*

[2] *Fraunhofer Institute for Digital Medicine MEVIS, Bremen, Germany*

**Editors:** Accepted for publication at MIDL 2026

## Abstract

Accurate pre-scan positioning in diagnostic imaging is essential for guiding acquisition and reducing manual calibration time, yet current automated approaches typically rely on dense volumetric representations that are not leveraging the geometric properties or sparsity of surface representations. In this work, we introduce a sparse, point-cloud–based framework for estimating patient-specific 3D locations and shapes of multiple internal organs directly from the body surface. Our method leverages a new dual-encoder PointTransformer architecture: one encoder processes a mean-shape point cloud comprising 20 anatomical structures, while a second encoder extracts features from the patient's body-surface point cloud. A shared decoder then predicts a deformed shape estimating the hidden individual anatomy patient. This enables accurate organ localization without volumetric rasterization or autoencoder-style bottlenecks. Trained on the German National Cohort (NAKO) dataset, our model substantially outperforms volumetric convolutional autoencoder (CAE) baselines, achieving a mean Chamfer Distance less than 5 mm and markedly lower surface-distance errors. These results demonstrate that sparse geometric learning with deformable point-cloud priors offers an efficient and highly effective alternative improving over dense convolutional deep learning methods for automated imaging workflow optimization.

**Keywords:** Point Cloud Processing, PointTransformer, Medical Image Segmentation, Patient Positioning

## 1. Introduction

In diagnostic imaging, the quality of the image is significantly influenced by the steps taken before the primary scan. These preliminary steps, also known as the pre-scan, encompass patient preparation and the planning of scan-specific settings. This process is vital to workflow, as it help standardize image acquisition and improve image quality by reducing artifacts and variability (Allen et al., 2023). For primary image acquisition, the patient is manually positioned on the table, then scout imaging is performed to quickly plan the detailed geometric layout, but the process can be influenced by the operator's skill and judgment (Van Rooyen and Pitcher, 2020; Al-Hayek et al., 2022). In certain cases, scout scans must be repeated because the patient was not positioned correctly. These scout images are used to visually verify the patient's correct position and to plan the geometry of the diagnostic sequences, including slice orientation, field-of-view, and scan range (Koken

et al., 2009). Fig. 1 captures the overall process of pre-scan steps. As the demand for radiology examinations rises, the efficiency of the preparatory steps becomes increasingly important to enhance patient throughput. Automated patient positioning systems are designed to streamline radiology workflows by identifying and localizing anatomical regions of interest and adjusting the patient table accordingly. Research supports the potential of these systems to reduce manual adjustments, minimize positioning errors, and enhance efficiency (Obuchowicz et al., 2024; Ghesu et al., 2022).

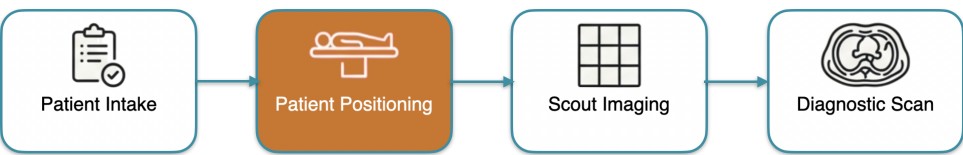

Figure 1: Diagnostic Imaging Preliminary Steps

For automated patient positioning, learning-based frameworks have emerged as the preferred choice due to their scalability and ability to generalize well to unseen cases. A comparison between manual positioning and artificial intelligence-based automatic positioning in CT imaging by utilizing RGB-D camera-based system is conducted, revealing that AI-based positioning significantly reduced the time required by 28% (Gang et al., 2020). The experiment is carried out by mounting RGB-D cameras above the scanner to detect key anatomical landmarks on the patient's body surface, enabling automatic table movement and centering. Similarly, Booij et al. created a 3D camera system that automatically identifies a patient's body shape to adjust the CT table height and centering, greatly enhancing positioning accuracy over manual techniques (Booij et al., 2019, 2021). The RGB-D camera's depth sensing capability generates a surface map of the patient's body. By utilizing anatomical landmarks, the internal anatomical regions of interest can be inferred.

Medical image analysis has predominantly relied on dense voxel grid representations and traditional machine learning approaches specifically designed for data structure on a grid. While advancements in dense 3D image analysis have been made, the limitations of structured data persist, including high computational demands and the need for large labeled datasets. On the other hand, sparse data representations like unordered point clouds offer a lightweight, topology-independent spatial sampling set that maintains millimeter-level detail while circumventing the memory overhead associated with meshes or dense grids. (Zhang et al., 2025; Heinrich, 2024; Bronstein et al., 2017). While the unordered nature of point clouds is encouraging because it offers lightweight solutions, it presents a challenge in incorporating local geometry and preserving permutation invariance.

## 1.1. Previous Works

In recent years, advancements in deep learning have enabled improvements in techniques for locating anatomical landmarks. Noothout et al. introduced a deep learning-based landmark localization technique utilizing fully convolutional neural networks (FCNNs) (Noothout et al., 2020). Zhao et al. developed a multi-resolution region proposal and segmentation network for orientation detection and localization, outperforming previous methods in accuracy and robustness (Zhao et al., 2022). Alansary et al. employed Deep Reinforcement

learning (DQN) to enhance landmark localization capabilities (Alansary et al., 2019). Kats et al. proposed a workflow that accurately predicts the coordinates of patients' internal organs and bones using depth images. They assert that this approach provides a contactless, fast, and standardized method for patient localization (Kats et al., 2025).

With their advantages, point clouds are becoming preferable choice in medical imaging, particularly when compared to volumetric 3D CNNs and vision transformers. Yu et. al. investigated the potential of point clouds in medical field, specifically in disease detection and treatment by developing attention-based point transformer model called 3DMedPt(Yu et al., 2021). Adams et. al. used unordered point clouds to learn Statistical Shape Modeling which is used in investigating and quantifying anatomical variations within populations of anatomies (Adams and Elhabian, 2023). In the work of Keller et. al., they train an implicit function to infer the 3D location of three important anatomic tissues from body shape model (Keller et al., 2024). Although HIT can learn to reconstruct soft tissue from the body surface, its precision in pinpointing the exact location of tissues within the body remains limited. The HIT model currently predicts the volume percentage of tissues but fails to accurately determine their precise pixel-wise location, particularly for sparse structures such as intra-muscular and visceral adipose tissue (IMVAT). Therefore, achieving accurate localization of these tissues continues to be a significant challenge.

## 1.2. Contribution and Outline

In this work, we propose a learning-based shape deformation model, DeformingPointTransformer, that utilizes body contours and a template shape to deform into a patients' internal organs and bones and determine their positions to eliminate the need for manual positioning. The model uses self attention modules to construct two encoders and a decoder - that employ the patient's body contour along with a population-based deformable template. The model can process these two inputs to accurately estimate the 3D shape of the internal structures.

To achieve strong performance across various anatomical variations and body types, we train our model using a comprehensive dataset of 6231 whole-body MRI scans from the German National Cohort (NAKO) study (Peters et al., 2022). This dataset encompasses subjects of different ages, body sizes, and physiological traits, allowing the model to generalize well across diverse patient groups. Our experimental results show the effectiveness of our proposed method in precisely identifying 20 different internal anatomical structures, such as bones and soft tissue organs, utilizing only depth images.

The key contributions of this work are as follows:

1. We show that a point cloud representation of body surface can be used to estimate the 3D positions of internal anatomical structures accurately and reliably.

2. We propose a transformer-based architecture, namely DeformingPointTransformer, which generates the 3D coordinates of the internal organs and bones using the point clouds of body surface and the mean deformable shape.

3. We demonstrate that in 3D generative tasks, point clouds are strong candidate to improve the workflow and process.

https://github.com/multimodallearning/DeformingPointTransformer

## 2. Materials and Methods

Our research utilizes 6231 complete MRI scans from the NAKO dataset (Peters et al., 2022). This extensive collection of images enables us to represent anatomical details across a broad spectrum of anatomical variations within the population.

### 2.1. Dataset Preparation

The dataset is composed of 3 main point cloud components; the body contour, deformable template, target internal anatomical structures. The preparation of the data and generation of the point clouds are comprised of 3 steps as demonstrated in Fig. 2. These steps are introduced in the subsequent sections.

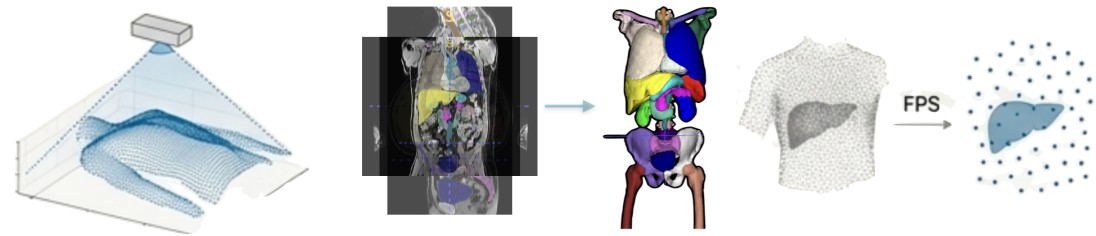

(*a*) Body Surface Extraction  (*b*) Boundary Extraction  (*c*) Point Cloud Generation

Figure 2: Three-stage dataset generation pipeline: (a) extraction of the body surface by simulating a top-down RGB-depth sensor from MRI volumes, (b) boundary extraction of 20 consistently present internal organs from automated segmentation masks, and (c) point-cloud generation of body surface, organ boundaries, and the deformable mean template using Farthest Point Sampling (FPS). Each training sample thus contains a body-surface point cloud and corresponding organ point clouds. In addition a deformable template is created by computing the mean mask across subjects and sampling its surface.

### 2.1.1. Body Surface Extraction

In patient positioning, an automation system can utilize RGB-D cameras to scan the body surface. In our dataset, the body surface is extracted using MRI volumes by simulating the depth sensor. An RGB-D camera positioned on top of the patient, looking downward, can determine the body surface locations using a top-down approach. Similarly, we employed the MRI scans to mimic this approach and extracted the body surface.

### 2.1.2. Boundary of the Internal Structure Extraction

As MRI scans are available, the 3D segmentation masks of anatomical reference labels are generated using TotalSegmentatorMRI (D'Antonoli et al., 2024), MRSegmentator(Häntze et al., 2024), and TotalVibeSegmentator(Graf et al., 2024). The models are trained on large-scale MRI datasets and used to generate segmentation masks in our datasets. The combined

masks cover 137 unique anatomical structures, comprising various organs, blood vessels, bones, and tissue types. We then applied the gradient operator to one-hot representation of each considered segmentation label in all three dimensions, took the absolute value, and summed the results across all dimensions to extract the boundary voxels. Although the resulting segmentation masks contain 137 anatomical structures, two criteria are set to determine the final selection of the labels. These criteria include the existence of the label in all masks and the sufficient number of non-zero boundary voxels across all subjects. Consequently, a final list of 20 internal structures are selected as shown in Table 1.

| Soft Tissue Organs | Bones |
| --- | --- |
| Lung Right | Hip Left |
| Lung Left | Hip Right |
| Liver | Clavicula Left |
| Kidney Right | Clavicula Right |
| Kidney Left | Femur Left |
| Pancrease | Femur Right |
| Duodenum | Sacrum |
| Aorta | Scapula Left |
| Heart | Scapula Right |
| | Vertebrae L4 |
| | Vertebrae L5 |

Table 1: The curated set of 20 internal anatomical structures used in this study. Labels were obtained from automated MRI segmentations (TotalSegmentatorMRI, MRSegmentator, TotalVibeSegmentator) and selected based on consistent presence across subjects and sufficient boundary-voxel density.

### 2.1.3. Point Cloud Generation

We used the same approach to generate the point clouds of the body contours and internal anatomical structure. After the body surface is extracted in 3D grid, we located the points and employed Farthest Point Sampling (FPS) algorithm to generate a point cloud with the size of $(16384, 3)$.

In generating a point cloud for each label, we created two versions: one with 1024 points and the other with 4096 points per organ. This step is crucial to assess the model's ability to handle a limited number of points. Similar to the body surface extraction process, the boundary voxels are detected, and for each organ, FPS operation took place to generate the point clouds with the size of $(1024/4096, 3)$. The resulting dataset comprises 6231 point clouds representing the body surface and 20 internal structures. Out of these pairs, 4780 are utilized for training, while 1454 are reserved for testing.

Finally, a template shape is generated using mean mask. We calculated the mean of one-hot representation of all the segmentation mask. Similar to Sec. 2.1.3, the point cloud of the deformable mean shape is generated. A sample of point clouds (body surface, deformable template, target) used in our dataset is available in Fig. 3.

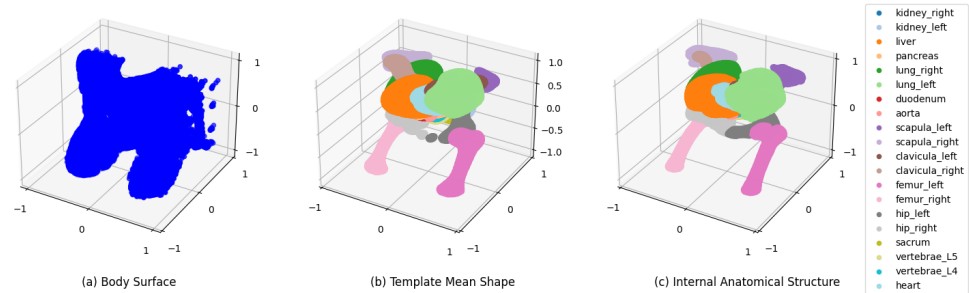

Figure 3: Point Cloud samples from the dataset.

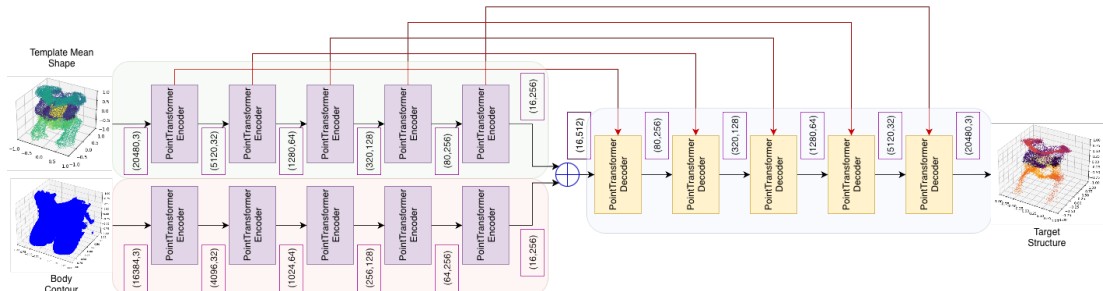

Figure 4: Overview of the proposed DeformingPointTransformer. A dual-encoder architecture processes the body-surface point cloud and the mean-organ template separately, fusing only their global features. The decoder applies topology-preserving skip connections solely from the mean-organ stream, enabling deformation of the template into patient-specific organ shapes conditioned on external body geometry.

## 2.2. Model Architecture

When processing 3D point clouds several challenges arise: the input data is unstructured and non-uniformly sampled - and hence a lack of a regular grid that would otherwise enable convolutions in voxel domains. Consequently, models are required that can operate on unordered sets and remain permutation invariant, while still being able to learn local, spatially dependent features. To construct a hierarchical feature representation that balances efficiency and accuracy makes methods for local neighbourhood aggregation necessary that can be applied during downsampling.

The PointTransformer architecture (Zhao et al., 2021) is well suited to address those challenges. It combines farthest-point sampling (FPS) with k-nearest neighbour (kNN) grouping to impose a meaningful local structure on the irregular input and efficiently deals with high-resolution point clouds. These steps together with grouped self-attention layers enable the network to learn localised geometric relationships to compute context-aware features while preserving permutation invariance.

In this work, we extend the PointTransformer encoder-decoder design to work on two related point clouds - one from the body surface and the other from the internal mean organ

model - jointly. This offers an effective balance between global shape learning and local deformation to infer the non-observed organs. During upsampling, trilinear interpolation propagates features from coarse to fine resolutions and is guided by the topology of the known mean model. This enables smooth reconstructions of dense point sets despite non-uniform input sampling and accurate prediction of the deformed shapes.

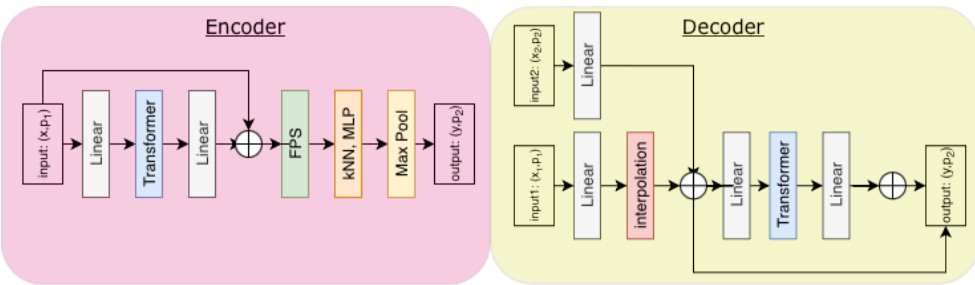

Figure 5: The details of Encoder and Decoder blocks. Each block follows the same architecture from PointTransformer.

The proposed DeformingPointTransformer architecture consists of two encoders and a single decoder, but the two encoded streams do not play symmetrical roles. The body-surface encoder extracts a global shape representation from the external point cloud, capturing coarse anthropometric variability. In parallel, the mean-organ encoder processes the template internal model and produces a hierarchical set of multi-scale features. Only the global feature vectors from both encoders are concatenated and passed to the decoder, ensuring that the predicted organ geometry is conditioned on the overall body shape.

Importantly, skip connections in the decoder operate exclusively on the multi-scale features of the mean-organ model stream. This design preserves the point ordering of the mean model throughout the decoding process and effectively turns the output into an ordered set aligned with the template topology. As a result, the network learns deformations of the internal anatomy rather than reconstructing an unordered point cloud from scratch. This structure enables the model to combine global shape cues from the body surface with local, topology-aware deformation fields derived from the mean internal model.

To process the mean shape and body contour point clouds, we employ five encoder blocks that utilize Transformer layers and down-sampling operations. Within each block, the number of points is reduced by a factor of four. At the latent code level, two point clouds with dimensions $[16, 256]$ are concatenated to form a single point cloud with dimensions $[16, 512]$. Subsequently, the decoder utilizes this latent code to generate the predicted organs, which have dimensions $[20480, 3]$.

In a hierarchical structure, incorporating a skip connection between the encoder and decoder would provide valuable information. However, in this task, the points derived from the body contours are not used in residual learning. Instead, they serve as a guide for deforming the mean shape. Consequently, we opted to utilize the points generated by the mean shape decoder and connect them to the decoder to enhance its output as they are the only branch where point ordering is stable. The model diagram is shown in Figure 4.

## 3. Results

We trained our model using 4780 randomly chosen point cloud pairs of body shape and their corresponding internal structures. For testing, we reserved a distinct set of 1454 samples. This sizable and diverse collection allows for a comprehensive evaluation of the model's performance and demonstrates its capacity to learn robust, generalizable features across a variety of anatomical variations.

The model is trained separately with both versions of the dataset. We denote the model trained on the 1024 points dataset DeformingPointTransformer1K and the other DeformingPointTransformer4K. Both models are trained with Adam optimizer with $10^{-3}$ initial learning rate. We decreased the learning rate in a multi-step fashion. Chamfer Distance (CD) cost function is used to calculate the loss value for the training. The CD is computed separately for each anatomical structure, to enable the model to optimally learn the localization and shape of each label.

To evaluate the model's performance, we used Chamfer Distance (CD), Average Symmetric Surface Distance (HD95) and Mean Absolute Detection Offset Error (DOE). CD validates the global structural correctness, and HD95 validates the local boundary accuracy. To calculate DOE, we derive the smallest bounding boxes that encapsulates the prediction and target point cloud and calculate the Detection Offset Error, defined as the absolute distance between the corresponding sides of the ground truth and predicted bounding boxes. To gain clearer insight into how the models behave across various spatial directions, the DOE is presented individually for each face of the bounding box (left, right, superior, inferior, posterior, and anterior). Breaking the results down this way lets us evaluate how well each approach places anatomical structures throughout the entire 3D space.

After training our own models, we compared them with the mean baseline model, and a Convolutional AutoEncoder adopted in the MONAI framework (Cardoso et al., 2022).

**Convolutional AutoEncoder**: As a volumetric baseline, we implemented a 3D convolutional autoencoder that operates on rasterized body-surface data. The input point cloud is first converted into a dense voxel grid using Gaussian splatting following (Heinrich et al., 2023), with a Gaussian kernel width of $\sigma = 1.7$ on a 2 mm isotropic grid. The resulting volume is processed by a 3D convolutional encoder–decoder architecture which incorporates a fully connected bottleneck to compress the encoded feature map into a 256-dimensional latent vector. The encoder uses seven 3D convolutional blocks with increasing channel widths and down-sampling strides, producing a compact bottleneck feature map that is flattened and projected to a 256-dimensional latent vector. The decoder expands this latent code back to the bottleneck shape via a fully connected layer and reconstructs the 21-channel volumetric output through a symmetric convolutional decoder. This baseline therefore evaluates a dense 3D rasterization and volumetric convolutional approach against our proposed sparse, point-based deformation model.

As further lower baseline **mean model** we simply evaluate the mean template shape (same "prediction" for all patients) against the ground truth for comparison.

In Fig. 6, we demonstrate that performance of the models on the Average Symmetric Surface Distance metric calculated over the test set. The metric is calculated for each test subject by first converting each point cloud to metric space (mm), then calculating the symmetric Hausdorff Distance 95th percentile.

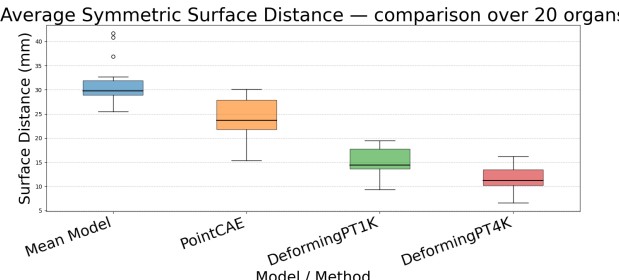

Figure 6: Average Symmetric Surface Distance. Results are computed over the test set.

| Metric | Mean Model | PointCAE | DPT-1K | DPT-4K |
|---|---|---|---|---|
| **CD** (mm) | $11.20 \pm 1.26$ | $8.07 \pm 1.01$ | $6.29 \pm 1.47$ | $\mathbf{4.69 \pm 1.19}$ |
| **HD95** (mm) | $31.07 \pm 4.24$ | $24.25 \pm 3.88$ | $15.03 \pm 3.02$ | $\mathbf{11.48 \pm 2.72}$ |

Table 2: **Aggregate Performance Comparison.** Mean and Std. Dev. ($\pm$) calculated across all 20 organs for each model. Lower values indicate better performance.

Similarly the plot in Fig. 7(a), we demonstrate that performance of the models on the Average Symmetric Chamfer Distance metric calculated over the test set. The metric is calculated for each test subject similar manner. Each predicted point cloud is first converted into metric space (mm), then the symmetric Chamfer Distance is calculated. Both HD95 and CD results suggests that the proposed structure can achieve outstanding performance in estimating the shape of individual structure and it can outperform convolutional based method. The plot in Fig. 7(b) demonstrate that the model made significant improvement compared to mean model.

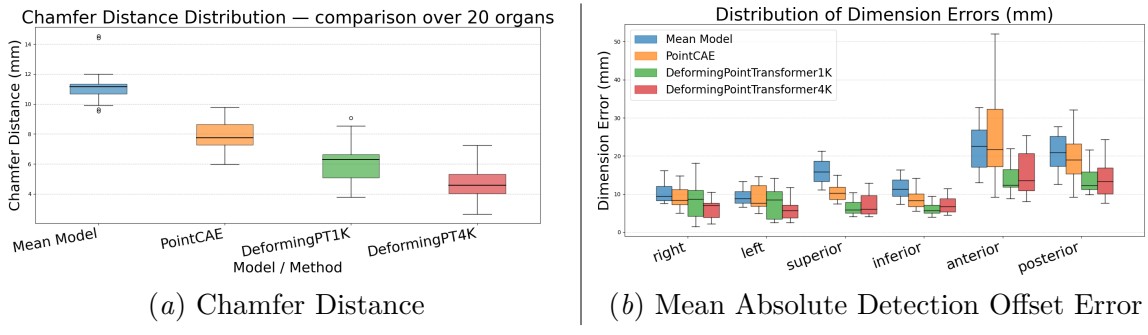

(a) Chamfer Distance           (b) Mean Absolute Detection Offset Error

Figure 7: Quantitative results computed over the test set. (a) Chamfer Distance. (b) Dimension offset error.

The more details of each metric for every label can be found in Appendix B. The experimental results validate the robust performance of the DeformingPointTransformer in localizing internal anatomy. The attention-driven architecture, which explicitly models

template deformation, proves significantly more effective than conventional ML baselines. Specifically, the attention mechanism captures the complex, non-linear spatial dependencies between external surface geometry and internal anatomical landmarks. Furthermore, the use of a deformable template imposes a strong geometric prior, ensuring that predicted organs maintain anatomical plausibility even when inferred from sparse surface data.

## 4. Discussion & Conclusion

The pre-scan process, to mitigate the throughput bottlenecks caused by error-prone manual positioning, automated systems were developed to localize anatomy and adjust the patient table, reducing preparation time. Although there have been attempts to use learning-based methods to automate this process, they often require dense data representations like depth images or voxel grids. This approach suffers from high computational demands and significant memory overhead. A strong alternative which is both lightweight and efficient is sparse point clouds.

In this work, we demonstrated for the first time that a sparse point cloud representation of the body surface can be used to accurately and reliably estimate the 3D positions of multiple internal anatomical structures. We proposed DeformingPointTransformer, a PointTransformer based model for estimating inner body structures from external depth sensors. By jointly providing the body-surface point cloud and the mean-organ template as inputs, the model learns how external body shape constrains the configuration of internal anatomy. The DeformingPointTransformer leverages this pairing by using the body surface to supply a global shape descriptor, while the mean-organ stream provides multi-scale, topology-preserving features. Through skip connections that operate only on the ordered mean-shape features, the decoder deforms the template into patient-specific organ geometries conditioned solely on the external surface.

The proposed DeformingPointTransformer consistently outperformed volumetric convolutional baselines on the NAKO dataset, achieving more accurate alignment of patient-specific organ geometries to the mean template. These results highlight the advantages of using sparse point clouds combined with a dual-input template-guided approach for 3D organ prediction. While this architecture is particularly effective for estimating internal anatomical landmarks with relatively low inter-subject variability, further evaluation is needed to assess its performance on tasks with greater anatomical variation, or real rather than simulated surface information obtained directly from depth sensors.

In conclusion, we proposed a model that accurately localizes internal organs using only body shape contours and an initial mean shape. In the future, it is possible to use implicit neural representations to further enhance the prediction point clouds for dense organ segmentation or distance map predictions without limitations to a certain voxel resolution.

## 5. Compliance with ethical standards

The German National Cohort (NAKO) study is performed with the approval of the relevant ethics committees, and is in accordance with national law and with the Declaration of Helsinki of 1975 (in the current, revised version).

## Acknowledgments

We gratefully acknowledge the financial support by German Research Foundation: DFG, HE 7364/10-1, project number 500498869.

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

## Appendix A. Table of Results

| Organ | CD | HD95 | Right | Left | Superior | Inferior | Anterior | Posterior |
|---|---|---|---|---|---|---|---|---|
| aorta | 6.30 mm | 19.42 mm | 22.05 mm | 8.04 mm | 13.15 mm | 11.45 mm | 26.09 mm | 23.73 mm |
| clavicula_left | 2.56 mm | 6.10 mm | 3.61 mm | 3.11 mm | 4.87 mm | 6.01 mm | 14.08 mm | 8.87 mm |
| clavicula_right | 2.47 mm | 5.60 mm | 2.68 mm | 2.63 mm | 4.21 mm | 6.41 mm | 8.02 mm | 11.24 mm |
| duodenum | 4.61 mm | 12.29 mm | 7.86 mm | 6.88 mm | 7.90 mm | 7.21 mm | 12.23 mm | 17.50 mm |
| femur_left | 4.57 mm | 11.22 mm | 2.07 mm | 4.38 mm | 4.96 mm | 5.91 mm | 10.24 mm | 9.03 mm |
| femur_right | 4.57 mm | 10.39 mm | 1.71 mm | 5.09 mm | 4.78 mm | 5.34 mm | 8.14 mm | 11.63 mm |
| heart | 7.75 mm | 17.00 mm | 16.66 mm | 11.75 mm | 15.04 mm | 14.44 mm | 39.47 mm | 26.19 mm |
| hip_left | 4.46 mm | 10.18 mm | 4.09 mm | 4.12 mm | 4.63 mm | 5.37 mm | 10.84 mm | 12.17 mm |
| hip_right | 4.51 mm | 10.11 mm | 3.82 mm | 3.80 mm | 4.77 mm | 5.62 mm | 11.75 mm | 11.03 mm |
| kidney_left | 4.60 mm | 13.03 mm | 7.99 mm | 9.78 mm | 9.05 mm | 7.61 mm | 15.76 mm | 16.08 mm |
| kidney_right | 5.21 mm | 15.07 mm | 7.93 mm | 11.70 mm | 10.87 mm | 8.32 mm | 15.08 mm | 15.52 mm |
| liver | 6.74 mm | 16.60 mm | 9.75 mm | 7.16 mm | 5.55 mm | 6.47 mm | 35.50 mm | 10.64 mm |
| lung_left | 5.97 mm | 14.64 mm | 9.86 mm | 3.63 mm | 5.02 mm | 5.03 mm | 10.96 mm | 18.98 mm |
| lung_right | 6.14 mm | 14.68 mm | 12.90 mm | 3.45 mm | 5.31 mm | 5.28 mm | 17.41 mm | 12.46 mm |
| pancreas | 4.88 mm | 11.97 mm | 6.71 mm | 7.10 mm | 6.48 mm | 12.85 mm | 25.64 mm | 13.71 mm |
| sacrum | 5.01 mm | 13.61 mm | 8.21 mm | 12.96 mm | 18.60 mm | 12.87 mm | 18.28 mm | 17.60 mm |
| scapula_left | 3.39 mm | 7.70 mm | 3.96 mm | 2.46 mm | 4.49 mm | 4.11 mm | 11.70 mm | 11.08 mm |
| scapula_right | 3.33 mm | 7.76 mm | 6.28 mm | 2.79 mm | 4.20 mm | 3.81 mm | 12.11 mm | 9.02 mm |
| vertebrae_L4 | 4.31 mm | 12.44 mm | 11.36 mm | 9.78 mm | 13.48 mm | 12.33 mm | 18.90 mm | 15.56 mm |
| vertebrae_L5 | 4.13 mm | 11.39 mm | 9.39 mm | 7.61 mm | 13.58 mm | 12.29 mm | 16.20 mm | 15.84 mm |

Table 3: The details of the performance of DeformingPointTransformer4K model

## Appendix B. Bar Plots of CD and HD95 Results

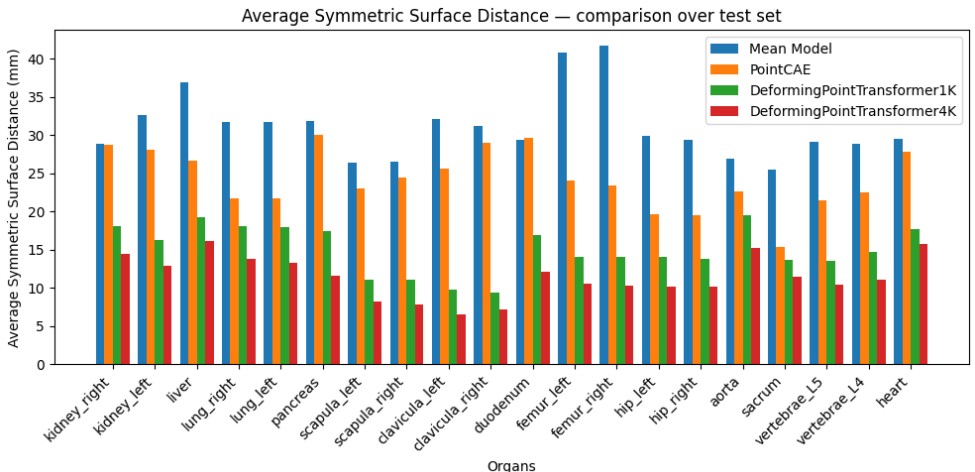

Figure 8: Average Symmetric Surface Distance

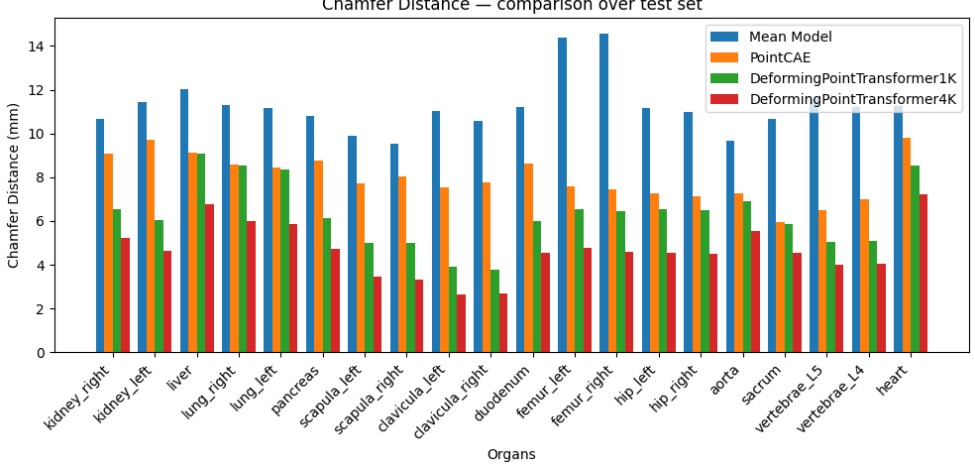

Figure 9: Chamfer Distance

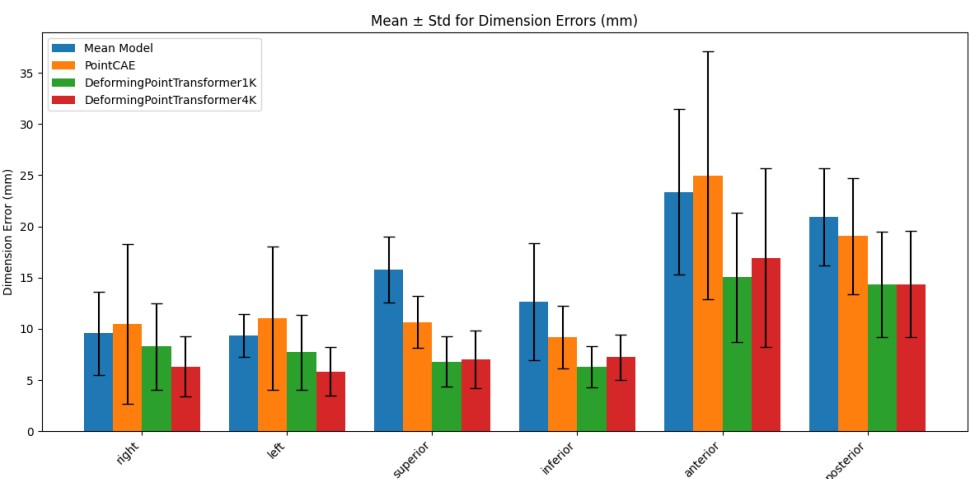

Figure 10: Mean Absolute Detection Offset Error

