# OpenReview forum: "From Surface to Viscera: 3D Estimation of Internal Anatomy from Body Surface Point Clouds"
_MIDL.io/2026/Conference — MIDL 2026 Poster_

### Official Review · Reviewer_FaSq · 2026-01-02

**Confidence:** 4
**Preliminary Rating:** 3
**Final Rating:** 4

**Summary:**

Motivated by the need for automated pre-scan positioning in diagnostic imaging, this paper proposes a method for estimating the 3D locations and shapes of internal organs from body surface point clouds. The authors propose to make use of a dual-encoder architecture where one branch processes a mean-shape template of 20 anatomical structures, and another encoder extracts features from the patient's body surface point cloud. The extracted features are combined and passed to a decoder that deforms the mean template into patient-specific organ geometries. The network is trained on a large dataset (6,231 whole-body MRI scans from the German National Cohort) and demonstrates strong performance.

**Strengths:**

The paper is generally well-written and easy to understand. I'll list its strengths in no particular order:
- The problem of estimating internal anatomy from external body surface for automated patient positioning is interesting and could improve radiology workflow efficiency.
- The asymmetric dual-encoder design with skip connections only from the mean-organ stream is interesting and seems to fit the task.
- The model is trained on a relatively large dataset with diverse anatomical variations (6,231 MRI scans from the German National Cohort), supporting the generalization claims.
- The model outperforms the baselines (the mean template and a convolutional autoencoder) by quite some margin.

**Weaknesses:**

While the paper addresses an interesting task, it has several weaknesses:
- Substantial parts of the paper focus on describing the problem of automatic patient positioning and why this is potentially interesting and relevant. The authors however never demonstrate if automatic patient positioning based on their estimations is actually helpful and don't even report any runtimes etc. that would at least help in estimating the usefulness and applicability of the method.
- The only learning-based baseline is a convolutional autoencoder. Comparisons with other point cloud methods (e.g., PointNet++, or a standard PointTransformer without the dual-encoder design) are missing.
- The authors motivate their choice of working with sparse point cloud representations with the computational demands and memory overhead of 3D grids and the models that process them. The paper however completely misses any actual comparison of e.g. processing time, required memory, etc.
- The body surface is simulated from MRI rather than acquired from actual RGB-D cameras. The paper acknowledges this limitation but doesn't provide any validation that the method would transfer to real depth sensor data with noise, occlusions, and different characteristics. I also don't see that the authors did an effort to simulate or account for e.g. noisy data or occlusions that might occur in a real scenario.

**Detailed Comments:**

- Some organs show much higher errors (e.g., heart). Is there an explanation of why certain structures are harder and can you give some comments on how this might impact clinical utility.
- Please introduce all abbreviations (e.g. HIT is missing) and always reference figures from the text to give the reader a hint when to look at them. Some figures just exist (e.g. Figure 5) without ever being mentioned or discussed.
- When describing the train/test split, the number seem to not really match. The dataset consists of $6231$ point clouds, where $4780$ are in training and $1454$ are in the test set. These number however sum up to $6234\neq6231$. Please correct this.
- The described learning-rate schedule remains unclear, as the authors only state that the learning rate is reduced in "a multi-step fashion".
- Please cite the Adam paper when using their optimizer.

**Justification Of Final Rating:**

The authors' rebuttal addressed most of my concerns and the paper substantially improved. I can now recommend to accept it. However, since I believe it would benefit from additional work (such as demonstrating concrete improvements to radiology workflows) I am rating it as a weak accept.

**Justification Of The Preliminary Rating:**

While I see that the task tackled in this paper is interesting and can potentially improve radiology workflows, there are substantial weaknesses that should be addressed before the paper can be accepted. I really miss a justification for working with point cloud that goes beyond stating they are more efficient (e.g. by actually comparing this with other architectures/ representations). As I'm also not fully convinced by the experimental design (especially the comparing methods), I would rate this paper as borderline.

**Questions To Address In The Rebuttal:**

I'd like the authors to focus on addressing my comments mentioned in the weaknesses/ detailed comments section. My main concerns are:
-  the lack of proving that the method is actually applicable for automatic patient positioning (as I understood the main motivation for this paper)
- the motivation for using point clouds relies on efficiency arguments that are never validated empirically (no runtime or memory comparisons)

Please also correct the smaller mistakes (e.g. the sample numbers) for a final version.

---

> ### Author Response · Authors · 2026-01-24
>
> Q: The authors however never demonstrate if automatic patient positioning based on their estimations is actually helpful and don't even report any runtimes etc. that would at least help in estimating the usefulness and applicability of the method.
> * Thank you for your concern. We have reported the inference latency and memory metrics in the revised manuscript. With an inference time of 97.15 ms per patient, we believe this approach offers a strong balance of accuracy and efficiency, making it highly applicable for clinical workflows. The results are included in [Section 3 / Table 4] of the revised manuscript.
>
> Q: The only learning-based baseline is a convolutional autoencoder. Comparisons with other point cloud methods (e.g., PointNet++, or a standard PointTransformer without the dual-encoder design) are missing.
> * We appreciate the opportunity to strengthen our evaluation. We have added a baseline comparison against a standard Point Transformer along with a convolutional decoder(without the mean-shape dual-encoder design). The results show that our proposed method, leveraging the strong geometric prior from the dual-encoder, significantly outperforms the single-encoder baseline.
>
> Q: The authors motivate their choice of working with sparse point cloud representations with the computational demands and memory overhead of 3D grids and the models that process them. The paper however completely misses any actual comparison of e.g. processing time, required memory, etc.
> * We agree that a quantitative comparison of computational efficiency is necessary. We have evaluated our model using the Microsoft DeepSpeed Profiler and PyTorch benchmarking tools to ensure rigorous metric reporting and We  added a detailed comparison of FLOPs, parameters, inference latency, and peak memory usage in Table 4 of the revision.
>
> Q: The body surface is simulated from MRI rather than acquired from actual RGB-D cameras. The paper acknowledges this limitation but doesn't provide any validation that the method would transfer to real depth sensor data with noise, occlusions, and different characteristics. I also don't see that the authors did an effort to simulate or account for e.g. noisy data or occlusions that might occur in a real scenario.
> * Thank you for this insight. We agree that validating the method under realistic sensor conditions is required. In the revised version, we tested our model on data augmented with simulated sensor noise and partial occlusions (point dropout). The results demonstrate that the model effectively estimates internal anatomy even when the input data is noisy or incomplete.
>
> Q: Some organs show much higher errors (e.g., heart). Is there an explanation of why certain structures are harder and can you give some comments on how this might impact clinical utility.
> * The higher error in the heart may stem from its location relative to the body surface or its complex shape variability compared to more static organs. We observe that for specific structures, the correlation between the body surface and internal anatomy is weak. This inherent ambiguity limits the upper bound of performance, as the internal position cannot be deterministically inferred solely from surface data.
>
> Q: Please introduce all abbreviations (e.g. HIT is missing) and always reference figures from the text to give the reader a hint when to look at them. Some figures just exist (e.g. Figure 5) without ever being mentioned or discussed.
> * Thank you for pointing this out. We have defined all abbreviations (including HIT) upon first use and ensured all figures are referenced in the text. Figure 5 has been removed as it was not essential to the discussion.
>
> Q: When describing the train/test split, the number seem to not really match. Please correct this.
> * We have corrected the discrepancy in the reported train/test split numbers.
>
> Q: The described learning-rate schedule remains unclear, as the authors only state that the learning rate is reduced in "a multi-step fashion".
> * We have clarified the implementation details in the revision to explicitly state the parameters of the multi-step learning rate schedule.
>
> Q: Please cite the Adam paper when using their optimizer.
> * We have added the formal citation for the Adam optimizer.

---

> > ### Comment · Reviewer_FaSq · 2026-01-26
> >
> > I would first like to thank the authors for their detailed response and for taking my considerations into account. I appreciate the changes made to the manuscript, especially the added baseline comparison, simulated noise & occlusion experiments, as well as the runtime information. I believe that the manuscript substantially improved over the initial version and I'm happy to raise my score.
> >
> > P.S.: There is missing T at the end of page 10: "... inference latency by nearly 60%. his efficiency stems from ..."

---

> > > ### Author Response · Authors · 2026-01-27
> > >
> > > Thank you for your follow-up and positive reassessment. We are glad that the added experiments and runtime analysis addressed the concerns and improved the manuscript. The typographical error on page 10 will be corrected in the revised version.
> > > Thank you again for your careful review and constructive feedback.

---

### Official Review · Reviewer_ey3Y · 2026-01-09

**Confidence:** 4
**Preliminary Rating:** 4
**Final Rating:** 4

**Summary:**

This paper presents a shape-deformation model that uses information from the body surface to deform a template shape of inner organs and bones to eliminate the need for manual positioning after scout scans in imaging procedures. The network has a two encoder, one decoder architecture and uses as inputs the patient body contour and a population mean shape. Instead of voxel images or triangle meshes, the method solely operates on unordered points clouds as inputs and output. The method is trained and evaluated on an extensive dataset and compared against a volumetric CNN baseline, where it is reported to achieve considerably lower surface errors.

**Strengths:**

* An important, but generally overlooked problem of pre-scan positioning is addressed with a novel approach.
* Using a mean-shape template is an effective way of introducing a strong anatomical prior.
* The dual-encoder architecture effectively helps with anatomical plausibility of the deformed template.
* A large-scale dataset is used to train and evaluate the proposed method.
* The evaluation is clear and the experiments are appropriate to show the performance and benefits of the method.
* The paper is well-written and easy to follow.

**Weaknesses:**

* The authors do not really motivate why an unordered point cloud is used instead of a connected polygon mesh.
* While the evaluation dataset is extensive, the surface point clouds are only simulated from MRI. No real RGB-D sensor data is used throughout the paper and no robustness experiments (noise, occlusion, etc.) are conducted.
* The source code does not seem to be publicly available.

**Detailed Comments:**

* The abstract is very well written and on-point. However, this sentence is unclear: "A shared decoder then predicts a deformed shape estimating the hidden individual anatomy **patient**."
* For sentences like this "Alansary et al. employed Deep Reinforcement learning (DQN) to enhance landmark localization capabilities (Alansary et al., 2019)." the usage of `\citet` in the beginning of the sentence is advised.
* Last page is blank and can be removed.

**Justification Of Final Rating:**

The rebuttal addresses several of my concerns and the added robustness experiment is appreciated. However, the evaluation remains on synthetic data and the robustness study uses simulated occlusion. Some of the clarifications in the author's response did not translate to changes in the manuscript (mean-shape construction, data splitting). Thus, my rating remains "weak accept".

**Justification Of The Preliminary Rating:**

This paper addresses a relevant problem using a novel and technically sound approach, supported by a large-scale experimental evaluation. However, the study relies on simulated surface data and the lack of robustness analyses limit the conclusions that can be drawn about real-world deployment. Despite these shortcomings, the method is well motivated and the results are promising. Overall, I believe the paper makes a solid contribution and I would like to see it discussed at MIDL.

**Questions To Address In The Rebuttal:**

* What are the reasons for preferring unordered point clouds over connected surface meshes, given that mesh connectivity could eliminate the need for kNN neighborhood construction and might better preserve topology?
* Please clarify whether the mean-shape template as been computed exclusively from the training set to avoid any data leakage.
* How sensitive is the method to noise in the surface point cloud that might occur with real RGB-D sensors?
* Please clarify whether the random train/test split was done at the patient level, with no subject overlap between the two sets.

---

> ### Author Response · Authors · 2026-01-24
>
> Q: The authors do not really motivate why an unordered point cloud is used instead of a connected polygon mesh.
> * Thank you for this comment. While we acknowledge that connected polygon meshes provide explicit topology (eliminating the need for neighborhood search), we believe point clouds offer superior flexibility for medical anatomy. Unlike meshes, which are sensitive to topological errors and reconstruction artifacts, point clouds provide a raw, fidelity-focused representation. Furthermore, our dynamic kNN neighborhood construction allows the model to learn geometric relationships adaptively rather than relying on fixed mesh edges. Coupled with positional encodings, the Transformer effectively captures global shape priors without the constraints of a rigid mesh structure.
>
> Q: While the evaluation dataset is extensive, the surface point clouds are only simulated from MRI. No real RGB-D sensor data is used throughout the paper and no robustness experiments (noise, occlusion, etc.) are conducted.
> * We appreciate this valuable suggestion. We agree that verifying robustness is essential for generalizability to real-world sensors. In [Section 3 / Table 3] of the revised manuscript, we have added experiments simulating sensor noise and partial occlusions (dropout). These results demonstrate that our model maintains strong performance even under certain data perturbations.
>
> Q: The source code does not seem to be publicly available.
> * The source code has been made publicly available. A link to the repository is included in the revised manuscript.
>
> Q: The abstract is very well written and on-point. However, this sentence is unclear: "A shared decoder then predicts a deformed shape estimating the hidden individual anatomy patient."
> * Thank you for the positive feedback. We have clarified this sentence in the revision. The term “shared decoder” refers to the final block of the DeformingPointTransformer, which fuses features from the mean shape and the patient's body surface. It predicts the dense displacement field to deform the template, thereby estimating the patient-specific internal anatomical structures.
>
> Q: For sentences like this "Alansary et al. employed Deep Reinforcement learning (DQN) to enhance landmark localization capabilities (Alansary et al., 2019)." the usage of \citet in the beginning of the sentence is advised.
> * Thank you for the advice. We have corrected the citation format throughout the manuscript to use \citet where appropriate.
>
> Q: Last page is blank and can be removed.
> * Thank you. The blank page has been removed.
>
> Q: Please clarify whether the mean-shape template has been computed exclusively from the training set to avoid any data leakage.
> * We confirm that the mean-shape template was computed exclusively using the training set to prevent any data leakage.
>
> Q: How sensitive is the method to noise in the surface point cloud that might occur with real RGB-D sensors?
> * We evaluated the model under varying noise levels to simulate real sensor artifacts. The results, added to the revised manuscript in [Section 3 / Table 3], indicate that the method is robust and can endure significant noise in the input data without a drastic drop in performance.
>
> Q: Please clarify whether the random train/test split was done at the patient level, with no subject overlap between the two sets.
> * Yes, the split was performed at the patient level. There is no subject overlap between the training and testing sets.

---

### Official Review · Reviewer_ChXr · 2026-01-09

**Confidence:** 4
**Preliminary Rating:** 5
**Final Rating:** 5

**Summary:**

This paper proposes a sparse point-cloud–based framework for estimating patient-specific 3D organ locations and shapes directly from body-surface geometry. Using a dual-encoder PointTransformer architecture with a deformable shape prior, the method predicts internal anatomy without relying on dense volumetric representations. Experiments on the NAKO dataset show improved localization accuracy over volumetric convolutional autoencoder baselines, suggesting the effectiveness of sparse geometric modeling for imaging workflow optimization.

**Strengths:**

A key strength of this work lies in its use of sparse point-cloud representations, rather than dense volumetric data, to estimate patient-specific internal organ locations and shapes. Compared to volumetric representations, point clouds are computationally efficient, preserve geometric boundaries and structural features, and are less sensitive to noise and intensity-based variability. While point-cloud methods have been extensively studied in computer vision, this work effectively leverages their advantages in a medical imaging context, offering a solution that is both methodologically innovative and practically relevant.

**Weaknesses:**

1) The results suggest that point-cloud density may play an important role in estimation accuracy. It would be valuable if the authors could further discuss how point density influences performance and whether denser point clouds consistently lead to improved results.

2) The use of point-cloud representations is a central strength of the paper. A more explicit discussion of their advantages in terms of computational and memory efficiency compared to dense voxel-based methods would further strengthen the contribution.

**Detailed Comments:**

1) Since the point clouds are derived from segmentation masks, it would be helpful for the authors to discuss how segmentation errors might propagate through the proposed pipeline, as well as potential strategies for handling challenging scenarios such as high noise, low image quality, or limited training data.

2) Given the generality of the proposed framework, a discussion on its potential extension to other imaging modalities and application scenarios would be a welcome addition.

**Justification Of Final Rating:**

The rebuttal provides a detailed response/modification to the points raised in my review and addresses my concerns in a satisfactory manner. The additional explanations help clarify the methodology and evaluation. I therefore maintain my rate.

**Justification Of The Preliminary Rating:**

Although point-cloud methods have been extensively explored in computer vision, this work successfully adapts and integrates these techniques into a medical imaging setting, demonstrating both methodological novelty and practical relevance. The combination of a principled geometric formulation and strong empirical performance makes this a compelling contribution. I recommand acceptance based on the novelty of the proposed approach and the solid experimental validation.

**Questions To Address In The Rebuttal:**

Please see weaknesses and detailed comments.

---

> ### Author Response · Authors · 2026-01-24
>
> Q: It would be valuable if the authors could further discuss how point density influences performance and whether denser point clouds consistently lead to improved results.
> * Thank you for your valuable insights. We agree that point density plays a crucial role in estimation accuracy. While extremely dense point clouds could yield further performance gains, our experiments demonstrating the improvement from 1,024 to 4,096 points (Table 2) empirically validate the positive correlation between density and performance.
>
> Q: A more explicit discussion of their advantages in terms of computational and memory efficiency compared to dense voxel-based methods would further strengthen the contribution
> * Thank you for highlighting this. We agree that the efficiency of point cloud representation is a fundamental contribution of this work. Accordingly, we have added a detailed comparison of computational cost (FLOPs) and memory efficiency in [Section 3 / Table 4] of the revised manuscript. DeformingPT4K demonstrates superior computational efficiency, achieving the lowest complexity among all models with only 4.89 GFLOPs. Compared to the PTConv baseline, it significantly reduces resource requirements, cutting peak memory usage by over 60% and inference latency by nearly 60%.
>
> Q: It would be helpful for the authors to discuss how segmentation errors might propagate through the proposed pipeline, as well as potential strategies for handling challenging scenarios such as high noise, low image quality, or limited training data
> * We appreciate your advice regarding robustness against real-world challenges, such as high noise and low data quality. To evaluate the model’s capability under these conditions, we simulated noise injection and random occlusion (dropout). The model demonstrated robust performance under these perturbations, and these results have been included in [Section 3 / Table 3] of the revised manuscript.
>
> Q: Given the generality of the proposed framework, a discussion on its potential extension to other imaging modalities and application scenarios would be a welcome addition.
> * Thank you for this suggestion. We have incorporated a discussion regarding this topic into the 'Future Directions' section of our revised manuscript

---

### Author Rebuttal · Authors · 2026-01-24

**Rebuttal:**

This rebuttal addresses the reviewers' feedback by adding detailed computational benchmarks and stress-testing the model against sensor noise. We further validated our approach with a new baseline comparison. We also open-sourced the code, and polished the text to clarify the experimental setup.

**Supporting Material:**

/attachment/c7b96a0d3799896389c57213373ce636d3a77a48.pdf

---

### Meta-Review · Area_Chair_PsJg · 2026-02-06

**Recommendation:** Accept (Oral)
**Confidence:** 5

**Metareview:**

All reviewers recommended to accept, they found the method novel and results promising.

---

### Decision · Program_Chairs · 2026-02-13

Accept (Poster)